# Polylactide/Carbon Black Segregated Composites for 3D Printing of Conductive Products

**DOI:** 10.3390/polym14194022

**Published:** 2022-09-26

**Authors:** Olha Masiuchok, Maksym Iurzhenko, Roman Kolisnyk, Yevgen Mamunya, Marcin Godzierz, Valeriy Demchenko, Dmytro Yermolenko, Andriy Shadrin

**Affiliations:** 1Centre of Polymer and Carbon Materials, Polish Academy of Sciences, 34. M.C. Sklodowska St., 41-800 Zabrze, Poland; 2International Polish-Ukrainian Research Laboratory ADPOLCOM, Centre of Polymer and Carbon Materials, Polish Academy of Sciences, 34. M.C. Sklodowska St., 41-800 Zabrze, Poland; 3E.O. Paton Electric Welding Institute of the National Academy of Sciences of Ukraine, 11. Kazymyr Malevych St., 03680 Kyiv, Ukraine; 4International Polish-Ukrainian Research Laboratory ADPOLCOM, E.O. Paton Electric Welding Institute of the National Academy of Sciences of Ukraine, 11. Kazymyr Malevych St., 03680 Kyiv, Ukraine

**Keywords:** poly (lactic acid)/carbon black composites, segregated structure, FDM 3D printing, electrical conductivity, morphology, thermal behavior

## Abstract

One of the most important directions in the development of additive manufacturing or three-dimensional (3D) printing technologies is the creation of functional materials, which allow not only prototyping but also the manufacturing of products with functional properties. In this paper, poly-lactide acid (PLA) /carbon black (CB) composites with segregated (ordered) structure have been created. Computer simulation based on the Mamunya geometrical model showed that the CB content within *φ* = 2.5–5 vol.% in the polylactide matrix leads to the formation of a continuous electrically conductive phase with an increase of electrical conductivity *σ_dc_* above the percolation threshold. The simulation results were experimentally confirmed by optical microscopy and studies of the electrical conductivity of the composites. It was found that increasing CB content from *φ* = 1 vol.% to *φ* = 7 vol.% in the composites causes insignificant (due to the segregated structure) phase changes in the polylactide matrix and improves the thermal properties of composites. Electrically conductive filaments for Fused Deposition 3D Printing (FDM) were developed from PLA/CB composites and then 3D printed. A correlation between the electrical conductivity *σ_dc_* and the CB content *φ* for base composites, filaments produced from them, and final 3D samples, has been found. Conductivity varies within *σ_dc_* = 3.1·10^−11^ − 10·10^−3^ S/cm for the filaments and *σ_dc_* = 3.6·10^−11^ − 8.1·10^−4^ S/cm for the final 3D-products.

## 1. Introduction

The rapid growth rates of high-tech branches of the industry determine the necessity for creating new and improving existing materials and technologies for obtaining high-quality products in the shortest time and with minimal monetary investments. Additive manufacturing, better known as 3D printing, belong to the technologies that meet the specified requirements and have captured every field of human life: from bioprinting [1] to 3D printed housing for life on Mars. Indeed, 3D printing by Fused Deposition Modeling (FDM) is the most widespread technology of additive manufacturing, and at the same time one of the most exciting and promising directions for further development and improvement. This technology works on the principle of additive (layer-by-layer) surfacing (stacking) extruded molten thermoplastic polymer material in layers according to the digital 3D model of the product.

Despite the large number of polymeric thermoplastic materials that can be used for forming 3D products using this technology, poly(lactic acid) or polylactide (PLA) is the most widespread and available consumable due to its perfect suitability and non-toxicity. PLA is an ecology-friendly biodegradable and biocompatible thermoplastic aliphatic polyether; mainly it is a polymer of lactic acid [2]. PLA has high strength (56.9 MPa) and a very low shrinkage coefficient (between 0.2 and 3%) compared with other thermoplastic polymers of general application used for FDM 3D printing. That makes it possible to create products of complex geometry, providing their high strength and surface quality.

At the same time, a trend in the improvement of existing and the design of new thermoplastic materials with unique properties for 3D printing [3,4], including their engineering applications, is up-to-date. Generally, these properties are achieved by adding functional components, in particular fillers, into the polymer matrix and, as a result, obtaining polymer composites with new properties, which are considerably different from the properties of the basic polymers. Conductive materials are the most important type of 3D printing functional materials, which have many potential applications such as electrodes [5,6,7,8], electromagnetic shields [9], solvent and electronic sensors [10,11], and possible heaters [12]. The application of such functional composites for 3D printing of various devices is only limited by our needs and imagination. It could be expanded to any industrial or scientific usage.

A wide range of fillers for the creation of polymer composites based on PLA, such as carbon black, carbon nanotubes, and metal powders, have been studied in a variety of works [13,14,15,16,17,18,19]. Adding such fillers into the polymer matrix leads to the growth of conductivity of the final composites in accordance with the percolation theory [20], when the nearby filler’s critical concentration forms a conductive framework in the polymer volume. The dependence of the conductivity*σ* on the content of a filler *φ,* according to this theory, can be presented as follows:*σ* = *σ_m_*
*(φ − φ_c_)^t^*(1)
where *σ_m_* is conductivity parameter that is often associated with the conductivity value of the filler phase, *φ_c_* is percolation threshold at which a 3D conductive cluster is formed, providing sharp growth of the conductivity depending on the volume content of the conductive filler, *t* is critical index (universal constant close to 2).

The conductivity of the filament based on polymer composite is affected not only by the type of filler but also by the method of its formation. Pridoehl et al. [21] suggested a method of filament production when the conductive filler is applied on a surface of the ready filament. However, in this case, the conductive filler will be distributed irregularly in the final 3D printed product, thus its conductive properties can be insufficient. It should be also mentioned that the mechanical strength of units produced by 3D printing with filled composites is in most cases lower compared with units produced by 3D printing with basic polymers, or produced by conventional methods like injection molding or hot pressing [22].

The formation process of the conductive filament described by Leigh et. al. [11] is complicated, because of numerous stages and the usage of volatile flammable toxic substances. Considerable progress was achieved by Tirado-Garcia et. al. [23], who have obtained conductive filament based on polyetheretherketone (PEEK) filled with a mixture of multi-walled carbon nanotubes (MWCNT) and nano-plates of graphene using a twin-screw extruder with a unidirectional rotation of the screws. The obtained results demonstrated a high level of conductivity of filament with randomly (statistically) distributed filler, but the problem of economic efficiency of high-cost materials application at the industrial scale, remained unsolved. Finally, it should be mentioned that commercial conductive consuming materials for FDM 3D printing are practically absent on the market.

The formation of composites with segregated (ordered) structure, contrary to random, allows for sharply decreasing the content of the conductive filler. From one side, such a method of formation negatively affects the possibility to process the composite by extrusion or injection molding, but, from the other side, that makes it possible to reach the required rheological and conductive properties of the composites and to obtain conductive filaments for the FDM 3D printing [24,25,26,27]. Thus, it is highly interesting to develop a simple and quick method of formation of filament, based on segregated polymer composites with a high level of conductivity for the additive manufacturing of final conductive 3D products, which can be used as heating elements for the welding of thermoplastics [28] and to carry through complex scientific studies of them.

## 2. Materials and Methods

### 2.1. Basic Materials

Thermoplastic PLA biopolymer (powder with particles size *D* = 200–400 μm produced by the milling of the commercial filament from Monofilament company) was used as the polymer matrix for the formation of composites and filaments on their basis. Average molar mass of the PLA is *M_w_* = 274,000 g/mol, dispersion of molar mass *M_w_/M_n_* = 2.4, density *ρ* = 1.24 g/cm^3^.

Carbon black (CB) ENSACO 250G (Imerys S.A., Paris, France) with density *ρ* = 1.82 g/cm^3^ and the particles size 50–100 nm and their agglomerates size up to *d* ~10 μm was used as the filler for the formation of the conductive polymer composites. The results of the scanning electron microscopy demonstrate a high level of agglomeration of the CB nanoparticles (Figure 1a) and the presence of a branched porous structure in the agglomerated particles (Figure 1b).

### 2.2. Formation of the PLA/CB Composites

Specimens of the PLA/CB composites with the segregated structure were formed by the method of hot compaction (pressing), stages of which are presented in Figure 2.

The following four filler contents were selected for the experimental specimens: 1, 2.5, 5, and 7 vol.%. To form a composite, the volume fraction of the filler was converted into a mass fraction, which makes it possible to determine the content of the composite components by weighing. The following equation was used for the calculation of the mass *C_f_* fraction of the filler:(2)$$C_{f}=\frac{\varphi_{f}\rho_{f}}{\varphi_{f}\rho_{f}+\varphi_{p}\rho_{p}}$$
where *φ_f_* is volume fraction of the filler; *ρ_f_* is density of the filler, *φ_p_* = (1 − *φ_f_*) is volume fraction of the polymer, *ρ_p_* is density of the polymer.

Using the precise mechanical mixing of the calculated mass quantity of both components in a ceramic mortar, the homogeneous mixture of two powders (polymer and filler) was obtained. Since the particle size of the polymer *D* and filler *d* differed significantly (*D >> d*), the filler particles were distributed on the surface of the polymer particles. During hot pressing, this structure was maintained, the polymer particles were deformed and joined, forming a continuous polymer matrix while the filler particles were located at the boundaries between the polymer grains, forming a continuous conductive framework in the polymer matrix. In such a way, the so-called segregated structure of the conductive phase in the composite is formed.

Further, the obtained mechanical mixture was placed in the steel closed form heated up to 200 °C, which is higher than the melting point *T_m_* of PLA (according to the DSC results *T_m_* of PLA is in the range of 160–186 °C depending on the conditions of crystallization). The mixture was compacted for 5 min under temperature 200 °C and 20 MPa pressure with further forced air cooling to room temperature. The specimens were formed in the form of cylinders of 9 mm in diameter and 12–13 mm in length.

### 2.3. Formation of Filaments and FDM 3D Printing Process

Filaments were formed from the polymer composites by injection molding using the IIRT-M equipment. Five cylindrical specimens of the polymer composite obtained as described in Section 2.2 were stacked one on top of the other in a cylinder of the IIRT-M device and were kept there for 5 min at 200 °C. Then, melted specimens were pressed out under 5N force through the outlet nozzle of the die, forming the filament rod with 1.72 ± 0.04 mm in diameter.

The 3D printed product was obtained in the form of a single-layer plate, which was formed from the filament by laying it back and forth on the substrate with the filament tightly adjoining each other by the sides using a 3D Intelligent Pen (Sunlu, China) with the diameter of the nozzle 0.6 mm. For all the specimens, the 3D printing temperature was 200 °C and the speed was 80 mm/min.

### 2.4. Optical Microscopy

Morphology investigations of the composites were performed using an optical polarizing microscope Versamet-2 (UNITRON, New-York, NY, USA) in the transmission mode. Respective transversal and longitudinal sections up to 20 μm in thickness were prepared using microtome LKB. Images were produced by the digital camera Nikon D350 (NIKON Corp, Tokyo, Japan) incorporated into the microscope.

### 2.5. Scanning Electron Microscopy (SEM)

The microscopy studies of the filaments and 3D specimens with different contents of CB were fulfilled by the Scanning Electron Microscopy (SEM) method. SEM studies were performed using the Quanta 250 FEG (FEI Company, Fremont, CA, USA) high-resolution environmental scanning electron microscope, operated at 5 kV acceleration voltage. The specimens were analyzed without coating under a low vacuum using a secondary electron detector.

### 2.6. Differential Scanning Calorimetry (DSC)

The thermal properties of the composites were investigated by the DSC method using TA Instruments DSC Q2000 (New Castle, DE, USA), certified by the manufacturer according to ISO 9001:2000 international standard. The device was calibrated to calorimetry precision ±0.05%. Specimens of 10 mg mass were loaded to the measuring cell and the cycled (heating-cooling-heating from 10 to 200 °C with the heating rate of 20 °C/min) tests in the nitrogen atmosphere (flow rate was 50 mL/min) were performed. The degree of crystallinity (*X_c_*) was determined according to the Equation (3):(3)$$X_{c}=\frac{∆Hm-∆Hc\;}{\omega ∆Hm0}\;\times \;100\%$$
where ∆*H_m_* and ∆*H_c_* are the enthalpies of melting and cold crystallization, respectively. ω and ∆*H_m0_* are the weight fraction of PLA and melting enthalpy of 100% crystalline PLA, respectively. ∆*H_m0_* enthalpy of meting for 100% crystalline PLA was taken 93 J/g [29,30,31].

### 2.7. WAXS

The structural analysis of the PLA/CB filaments and 3D specimens was performed by wide-angle X-ray scattering (WAXS). A D8 advance diffractometer (BRUKER AXS, Bremen, Germany) with a copper anode (1.5418 Å) was used, operating at 40 kV and 40 mA. WAXS curves were recorded in a Bragg-Brentano geometry, over a diffraction angle (2θ) range: 2–50°, with the step size of 0.02° using a LYNXEYE XE-T detector.

### 2.8. Measurements of the Electrical Conductivity

The measurements of the electrical resistance (R) of the specimens were fulfilled under direct current using a two-electrodes scheme with a potentials difference of 10 V using terraohmmeter E6-13A (Radiotehnika, Riga, Latvia). Each specimen was placed between two nickel-bronze electrodes of 15 mm in diameter. For reliable contact, the applied pressure was 0.1 MPa. Further, the electrical resistance R (Ohm) was recalculated into conductivity *σ* (S/cm) using equation:(4)$$\sigma =\frac{h}{R\cdot S}$$
where *R*—electrical resistance, measured by terraohmmeter, *h*—thickness of the specimen; *S*—the square of the specimen. Three specimens of each composite were measured with further averaging of the results.

Cylindrical samples were placed between the electrodes. For filaments, the conductivity was measured along the 15 mm filament segment, which was clamped between two crocodile clips. For 3D printed specimens, the conductivity was measured along the filaments laid out side by side on the substrate.

### 2.9. Static Mechanical Testing

Mechanical tests at uniaxial tension mode were performed using FP-10 tension machine according to EN ISO 527-1:2017 and EN ISO 20752:2019 standards at 20 ± 2 °C with linear clamps velocity 5 mm/min. Maximum loading until the breaking of the specimens were measured during the tests, average strength F was calculated for at least three specimens of each composite.

### 2.10. Computer Modeling

To determine the spatial distribution of the particles of the conductive filler of various contents φ in the polymer matrix, as well as to determine the level of filler content in the composite required for the creation of conductive chain structures that act as conductive channels within the specimen, the computer modeling in accordance with the Mamunya geometrical model of the segregated system [32,33] have been performed. The model calculates an ordered structure in the shape of a three-dimensional framework of the filler particles (Figure 3).

## 3. Results

### 3.1. Modeling of the Segregated Structure of PLA/CB Composites

The computer models of the spatial distribution of CB at various concentrations *φ* (1, 2.5, 5 and 7 vol.%) with the size of nanoparticles ~50 nm and the size of their agglomerates up to *d* ~10 μm in the PLA matrix with the size of polymer grains of around 200 μm, are presented in Figure 4.

According to the results of computer modeling, CB particles are ordered in all polymer composites. The content of *φ* = 1 vol.% (Figure 4a) is not sufficient for the formation of a continuous conductive phase, since the particles are not in contact with each other, so the conductivity of such composite should be close to the pure polymer. By increasing the CB content up to *φ* = 2.5 vol.% (Figure 4b) the filler particles are still not in contact with each other, so the level of conductivity is slightly higher than in the previous case, but it is not yet sufficient. Polymer composite with *φ* = 5 vol.% (Figure 4c) contains the continuous conductive chains of the CB particles, and their aggregates are detected in some places on the borders between the polymer particles. The conductivity of such composites will grow up sharply compared with the two previous composites. Further growth of the CB aggregates is detected for polymer composites with *φ* = 7 vol.% (Figure 4d). Such aggregates merge with each other and form the continuous conductive cluster.

### 3.2. Morphology of the PLA/CB Composites, Filaments and 3D Specimens

The evaluation of the spatial distribution of the CB particles with their various contents φ (1, 2.5, 5, and 7 vol.%) in the created PLA/CB composites was analyzed using optical microscopy. Figure 5 presents images of the cross sections of composites produced by hot compaction.

The images depict the evolution of the CB conductive phase located on the borders between the PLA particles with the increase of its content. Due to the peculiarity of the specific formation of specimens, namely the compression of the polymer and filler mixture, the PLA particles become not spherical, but partly flattened. In the case of a low con- centration of the CB, its particles practically do not contact each other, remaining between the PLA grains that form the insulating polymer composite (Figure 5a). Composite PLA-2.5 vol.% CB shows the beginning of the formation of the conductive network of the filler (Figure 5b).

With an increase in the CB content to *φ* = 5 vol.%, a significant darkened boundary is observed between the PLA grains, filled with CB particles (Figure 5c). This indicates the formed conductive framework inside the polymer matrix, where most of the CB particles are in direct contact with each other. This effect is noticeably more intense for composites with CB content *φ* = 7 vol.% (Figure 5d), in which a considerable expansion of CB areas leads to a significant increase in conductivity.

Figure 6 represents images of the morphology in transversal cross sections of the filaments produced from the cylindrical specimens of PLA/CB composites. When forming the filament, a melted cylindrical specimen of the composite with a diameter of 9 mm is narrowed to a diameter of 1.72 mm in the outlet nozzle of the die. As can be seen from Figure 6, the segregated structure of the composite is maintained, however, the conductive network is extended along the melt flow, and the butts of the elongated conductive structure of CB can be seen in the microphoto. With an increase in the CB content, the diameter of the elongated structures increases, and their number also increases, which corresponds to the morphological structure of cylindrical samples (Figure 5).

The images of the longitudinal section of the filament are shown in Figure 7. When forming a filament from a two-component (polymer-filler) composite, the melted PLA particles elongate, forming polymer strands, between which carbon black particles are located in the form of aligned continuous chains (Figure 7a). Figure 7b demonstrates the structure of the conductive chain formed by CB particles.

Figure 8 shows cross-sectional SEM images of 3D printed specimens using a filament of the structure described above (see Figure 6 and Figure 7) with different CB content. It can be seen that the distribution of CB in the PLA matrix of 3D printed specimens correlates with its distribution in the filaments. It is easy to follow a clear pattern of expansion of the dark zones of the conductive filler in 3D specimens with an increase in the CB content.

### 3.3. Thermal Properties of the PLA/CB Composites, Filaments and 3D Specimens

The thermal behavior of unfilled PLA and PLA-based filaments with various content of CB filler and 3D printed specimens based on them were studied by DSC. Figure 9 presents the DSC curve for the first heating cycle (as it is, with thermal history that the polymer earned during synthesis and processing) of the unfilled PLA filament and the composite filaments containing 1, 2.5, 5, and 7 vol.% of CB. Three temperature transitions are observed on the curves: (1) is an endothermic peak associated with the glass transition; (2) is an exothermic peak of cold crystallization; and (3) is an endothermic peak of melting. The thermal parameters of these transitions, such as glass transition temperature (*T_g_*), temperatures of “cold crystallization” (*T_cc_*), melting temperatures (*T_m_*), “cold crystallization” enthalpy (∆*H_c_*), melting enthalpy (∆*H_m_*), and the degree of crystallinity (*X_c_*), are presented in Table 1.

The glass transition at ~67 °C is accompanied by a large endothermic enthalpy re-laxation peak. When, during the production of a filament, a polymer passes from a melt to a rubbery state upon cooling, and then is transformed in the glassy state through the glass transition temperature *T_g_*, it exists in a non-equilibrium state with an excess of such a thermodynamic parameter as enthalpy. In the case of slow cooling from temperatures above their *T_g_*, the polymer has a lower excess enthalpy, since its polymer chains have gained the ability to find a more energetically favorable orientation, which makes the polymer structure closer to thermodynamic equilibrium. In the case of rapid cooling, the polymer chains are “frozen” in high-energetic conformations and the polymer is in a non-equilibrium state [34,35].

It was shown in refs. [36,37,38,39] that keeping the polymer at a temperature *T_a_* (the so-called aging temperature) close to *T_g_*, it gradually passes into a more equilibrium state with a decrease in the excess enthalpy. This process is closely related to the change in the mechanical characteristics of PLA as strength and modulus increase, while toughness decreases drastically [37]. Barrasa et al. [38] found that long-term aging under natural conditions (up to 1 year) significantly affects the polymer structure and mechanical characteristics. In ref. [35], it is noted that one of the main disadvantages of PLA is physical aging, which leads to the instability of the structure and physical properties of the material. It has also been shown [38] that the extrusion temperature of the PLA filament affects the properties of the final product. Thus, the thermal and mechanical properties of the PLA filament depend on the regime of its formation and the temperature-time influence of the environment, i.e., on its thermal prehistory.

For PLA composites, the enthalpy of relaxation and physical properties depend on the influence of the components of the composite [40,41]. In our case, the intensity of the enthalpy relaxation peak is the maximal for pure PLA; moreover, this peak has a double structure. With an increase in the CB content in the composite, the intensity of the peak slightly decreases, which indicates the stabilizing effect of CB on the orientation of molecular chains. At the same time, the glass transition temperature *T_g_* remains constant for all PLA composites.

The presence of a large exothermic peak of cold crystallization indicates that during heating some part of the amorphous phase in the polymer is ordered and converted into a structured crystalline phase. As follows from the table, the cold crystallization peak temperature *T_cc_* increases with an increase in the CB content in the composites, which indicates that the reorientation of molecular chains in the presence of a filler is limited. The highest values of the heat of crystallization Δ*H_cc_* are observed for PLA-1 vol.% CB and PLA-2.5 vol.% CB composites.

During further heating the crystalline phase formed by cold crystallization, and also present in the primary PLA, melts and gives a large endothermic peak on the DSC curve. The melting temperature *T_m_* does not depend on the CB content, while the heat of melting Δ*H_m_* shows the highest values for composites containing 1 and 2.5 vol.%. The result of the calculation of the degree of crystallinity *X_c_* of virgin PLA filament according to Equation (3), as the difference between the heats of melting and crystallization, is presented in Table 1. As can be seen, the lowest degree of crystallinity is observed for the unfilled PLA filament, while the highest *X_c_* value is shown by the composite with 7 vol.% of CB. Apparently, CB particles serve as crystallization centers during the formation of the PLA structure under the conditions of the filament processing.

Figure 10 shows DSC curves (Ⅰ run) of the 3D-printed specimens, obtained from the composite filaments with various CB content. As follows from Table 1, the glass transition temperature *T_g_* ≈ 69 °C remains unchanged for all composites. The curves also show an endothermic enthalpy relaxation peak, the intensity of which is approximately the same for all composites. As for cold crystallization, the temperature of its exothermic peak *T_cc_* and the heat of crystallization Δ*H_cc_* have the highest values for pure PLA, while the opposite case was observed in filament samples. Additionally, the temperatures and heat of melting Tm and Δ*H_cc_* of pure PLA have the highest values. The degree of crystallinity of the 3D specimens increases with the content of CB in the composites, as well as in filament specimens, however it has higher values. These results testify to the influence of the molding conditions of the material on its thermal properties. The 3D printing mode, when the polymer melt flows from a nozzle with a diameter of 0.6 mm onto a solid substrate (having a temperature of 200 °C), forms the structure of a single-layer 3D printed specimen with a higher degree of crystallinity than the regime of extruding a filament from a nozzle with a diameter of 1.72 mm with cooling in the air. This result may be due to the additional orientation of molecular chains during the melt flow in a nozzle of a smaller diameter.

The results of testing 3D specimens during a second run, shown in Figure 11, are of interest. First, there is no enthalpy relaxation peak. Secondly, the glass transition temperature is ~5 °C lower than for the first run. Obviously, due to the elimination of thermal history during the first run, this sample has a more equilibrium state at a temperature below *T_g_*. The degree of crystallinity of the samples is approximately two times lower than that observed for the samples in the first run. The value of *X_c_* is determined by the ratio of the heat of melting Δ*H_m_* and crystallization Δ*H_cc_* for both test cases (I or II run). During the second run, the values of the heat of melting are only slightly higher than the value of Δ*H_m_* for the samples of the first run, while the values of the Δ*H_cc_* are noticeably higher. Obviously, for a more equilibrium structure, the reorientation of polymer chains during cold crystallization is facilitated.

### 3.4. WAXS

Figure 12 presents the WAXS patterns of the initial (pure) and the composite filaments with various CB content *φ* (1, 2.5, 5 and 7 vol.%). Figure 13 presents WAXS patterns of the 3D specimens, obtained from them.

The particular diffraction peak located at approximately 26.5° is related to CB in all specimens besides the initial filament. Initial PLA filament showed that this material is characterized by an amorphous disordered structure, as evidenced by a wide halo without clear reflections with a maximum at 2θ = 16.5° (Figure 12). An increasing of the CB content in the composites up to 5–7 vol.% causes the appearance of low-intensity diffraction peaks around of 17° and 18.5°, that is related to the crystalline phase of PLA. Therewith, the PLA crystallinity increasing with the introduction of CB is a little more visible on the WAXS patterns of 3D specimens, which can be a result of the additional influence of the 3D printing process (Figure 13). Thus, the introduction of CB nanoparticles, which act as a nucleating agent to PLA matrix, can improve the crystalline structure of PLA but, due to the fact that the filler in the segregated structure of specimens is located between the polymer particles, this effect is negligible.

### 3.5. Electrical Conductivity of the Composites

The results of the electrical conductivity measurements for three types of composite specimens—cylindrical rods formed by hot pressing, filaments, and 3D printed specimens using these filaments, with various contents of CB, are presented on the graph describing the dependence of conductivity *σ_dc_* on the content of CB *φ* (Figure 14).

As can be seen, there is no pronounced percolation threshold in these dependences; the electrical conductivity of the composites smoothly increases with increasing CB concentration. However, its value can be estimated using the geometric Mamunya model [32,33], in which the percolation threshold value is related to the parameters of the conductive framework formed in the segregated system. According to the model, the value of the percolation threshold of the segregated conducting phase *φ_cs_* is defined as:(5)$$\varphi_{cs}=\varphi_{c}\left[1-{\left(1-\frac{nd}{D}\right)}^{3}\right]$$
where *φ_c_* is the percolation threshold for the statistical distribution of the filler, *n* is the number of filler layers in the wall of conductive framework, *d* is the size of the conductive particle, *D* is the size of the polymer particle. The following values of parameters were taken for calculation: *d* = 10 μm (size of CB agglomerates), *D* = 200 μm, *n* = 2 (if each polymer particle is covered with one layer of filler particles, then two layers are contained in the intergranular region during pressing filler particles). The value of *φ_c_* is taken from ref. [40], where a PLA-based composite with a homogeneous CB distribution was obtained using a torque rheometer. The percolation threshold determined by the authors was 8 wt.%, which, in accordance with Equation (2), is 5.5 vol.%. Calculation by Equation (5) gives the value of the percolation threshold φ_cs_ = 1.5 vol.% for the segregated system formed in this work.

It follows from Figure 14 that an increase in the concentration of CB in PLA from 1 to 2.5 vol.%, i.e., when passing through the percolation threshold, leads to an increase in conductivity in cylindrical samples and filaments by more than two decimal orders. For 3D printed samples, this dependence is weaker. Figure 5 (Section 3.2) shows the formation of a pronounced conductive framework at a CB content of 5 vol.%, and in accordance with this, the conductivity value increases and is in the range of 10^−3^–10^−5^ S/cm. In the case of 7 vol.% carbon black in the PLA matrix, the conductive network becomes more developed and the conductivity reaches values of 10^−2^–10^−3^ S/cm for different types of composites. The highest values of conductivity among all specimens were reached for the filament specimens having conductivity close to *σ_dc_* = 1.0·10^−2^ S/cm.

It should be noted that *σ_dc_* in the printed 3D specimens has a lower value compared with other composites. This can be explained by the fact that during the 3D printing process using filament with 1.72 mm in diameter the melted composite is squeezed out through a nozzle with a diameter of 0.6 mm. Apparently, the conductive structure formed in the filament is partially destroyed when the melt flows through the smaller nozzle. This results in a decrease in the conductivity value *σ_dc_*.

### 3.6. Mechanical Strength of the PLA/CB Filaments and 3D Specimens

The results of mechanical tests of the filaments and 3D specimens with various CB content *φ* are presented in Figure 15.

It is evident that the increase of CB content in the segregated PLA matrix decreases the mechanical strength of the filled filaments, as a result of the appearance of an inter-facial boundary, which is the place where defects are concentrated. As the filler concentration increases, local defects and stress concentrations play an increasingly important role in the mechanical properties of the composite, weakening interphase adhesion between the binding polymer material and CB agglomerates.

The strength values for 3D printed specimens depend mainly on the printing parameters, which affect the formation of the sample structure, weakening the interlayer adhesion and strength. The strength value of 3D specimens (*σ* = 34.2 MPa) is 41% lower comparing with the filament (*σ* = 57.8 MPa).

## 4. Conclusions

Optical and electron microscopy methods were used to study the structure of PLA-carbon black composites, which were processed in the form of block specimens (cylinders), from which filaments were prepared for 3D printing and 3D printed specimens. In block specimens, a segregated structure of the conductive filler was formed, which was retained during the production of filaments and 3D printing. The formation of an elongated PLA-CB structure during the flow of composites through a nozzle is shown.

A computer simulation of the segregated structure was carried out at different contents of the conductive filler, which showed good agreement with morphological images.

The study of the thermal properties of composites made it possible to establish patterns of the structure formation of filaments and 3D printed samples. The presence of an endothermic enthalpy relaxation peak in the region of the glass transition temperature was found, which is associated with the formation of non-equilibrium state polymer chains during the production of the filament and 3D printing. The elimination of the thermal history returns the polymer to a more equilibrium state, and the enthalpy relaxation peak disappears. The presence of cold crystallization in the temperature range of 100–140 °C was shown in all the studied samples. The influence of different CB concentrations on the thermal characteristics of the composites has been traced.

The study of the electrical conductivity of the composites depending on the CB concentration showed the absence of a pronounced percolation threshold. However, its value was estimated using the geometrical Mamunya model, which gave a low percolation threshold φ_cs_ = 1.5 vol.% for a segregated system. It is shown that with a CB content of 7 vol.%, the filament has the highest conductivity *σ_dc_* = 1.0·10^−2^ S/cm, while the 3D printed sample has the lowest conductivity due to the partial destroying of the conductive framework when passing through a small diameter nozzle. The mechanical strength of the filament and 3D printed material decreases monotonically with increasing carbon black content in the composites.

## Figures and Tables

**Figure 1 polymers-14-04022-f001:**
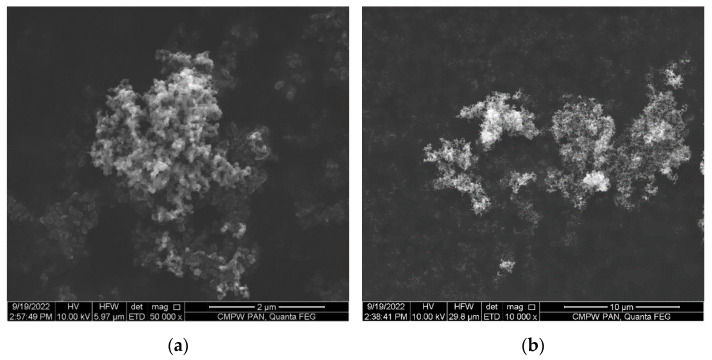
SEM images of the carbon black (CB) ENSACO 250G (**a**,**b**).

**Figure 2 polymers-14-04022-f002:**
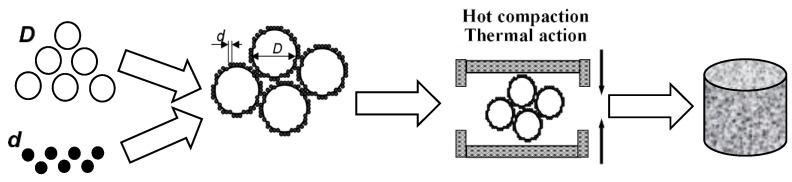
Scheme of the formation process of the segregated polymer composites by the method of hot compaction, where *d*—diameter of the CB particles, *D*—diameter of the PLA particles.

**Figure 3 polymers-14-04022-f003:**
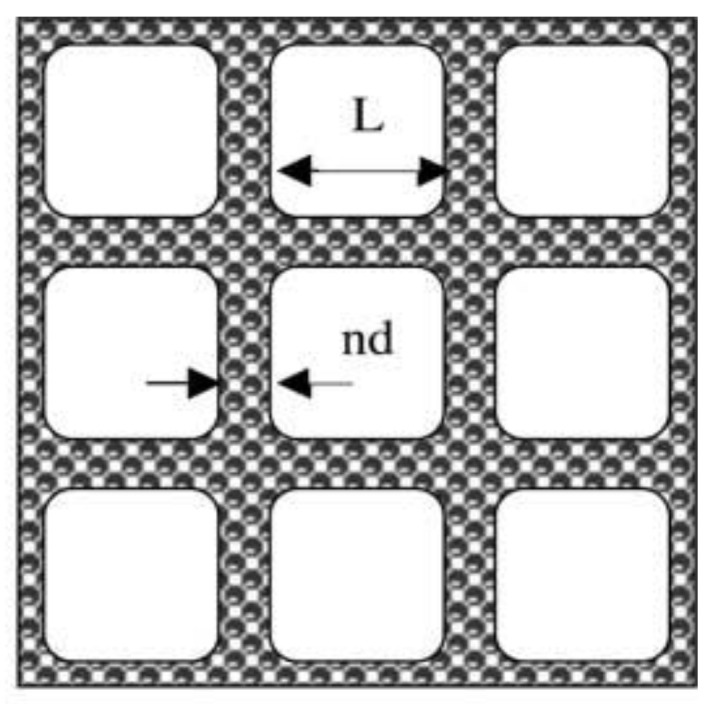
Mamunya geometrical model of the framework structure of the segregated polymer system. Reprinted from Ref. [32]. Model parameters: *D = L + nd*, where *L* is the size of an excluded volume of the polymer, *n* is the number of layers of the conductive particles in the framework, *D*—size of the polymer particles, *d*—size of the filler particles.

**Figure 4 polymers-14-04022-f004:**
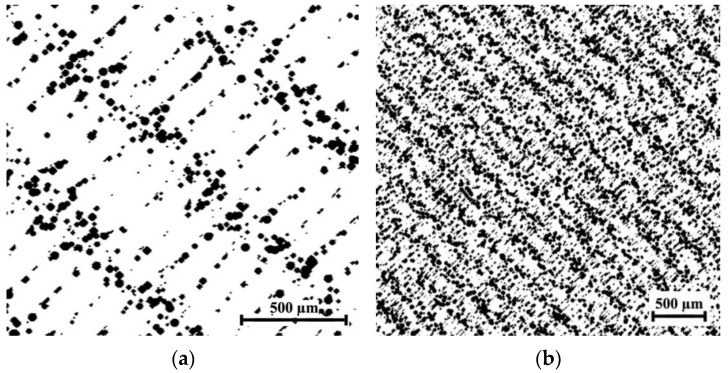
Results of computer modeling of the segregated structure of PLA/CB composite depending on evolution of CB content *φ*: 1 vol.% (**a**), 2.5 vol.% (**b**), 5 vol.% (**c**), and 7 vol.% (**d**).

**Figure 5 polymers-14-04022-f005:**
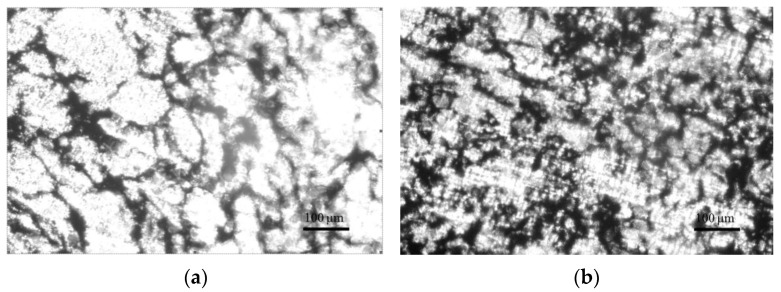
Optical images of the cross-sections of cylindrical specimens PLA + 1 vol.% CB (**a**). PLA + 2.5 vol.% CB (**b**), PLA + 5 vol.% CB (**c**), PLA + 7 vol.% CB (**d**).

**Figure 6 polymers-14-04022-f006:**
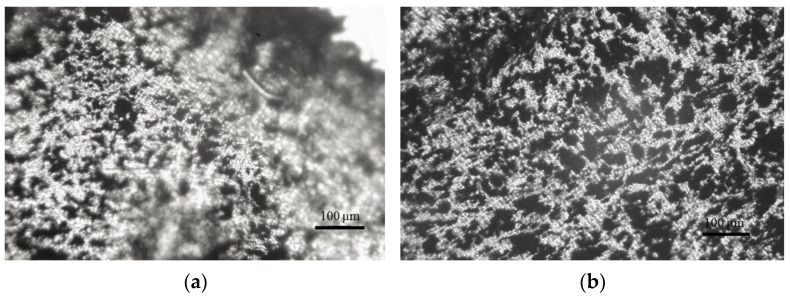
Optical images of the transversal cross-sections of the filaments PLA + 1 vol.% CB (**a**), PLA + 2.5 vol.% CB (**b**), PLA + 5 vol.% CB (**c**), PLA + 7 vol.% CB (**d**).

**Figure 7 polymers-14-04022-f007:**
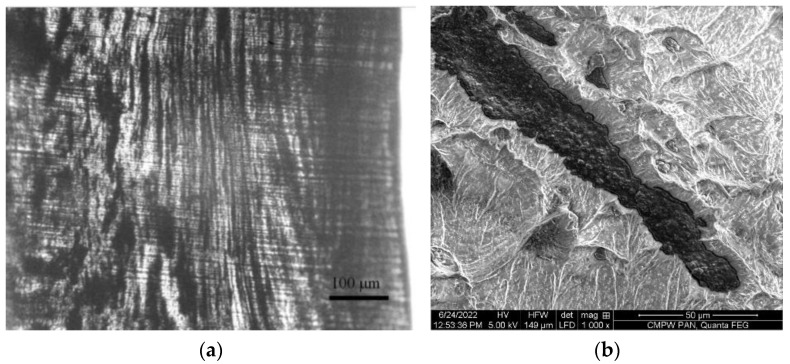
Images of the longitudinal section of the filament with the CB content *φ* = 2.5 vol%: (**a**) optical microscopy, (**b**) SEM.

**Figure 8 polymers-14-04022-f008:**
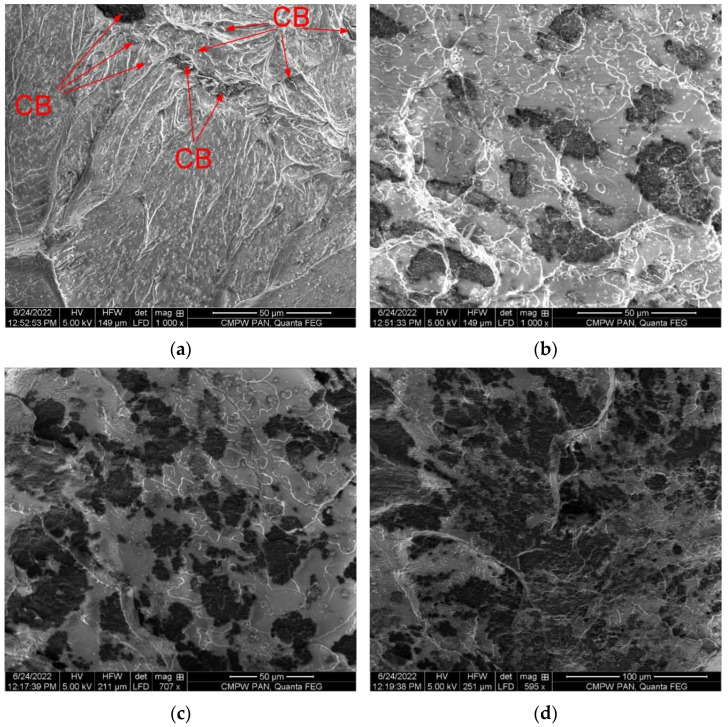
SEM images of the transverse cross-section of the specimen’s 3D printed with filaments: PLA + 1 vol.% CB (**a**), PLA + 2.5 vol.% CB (**b**), PLA + 5 vol.% CB (**c**), PLA + 7 vol.% CB (**d**).

**Figure 9 polymers-14-04022-f009:**
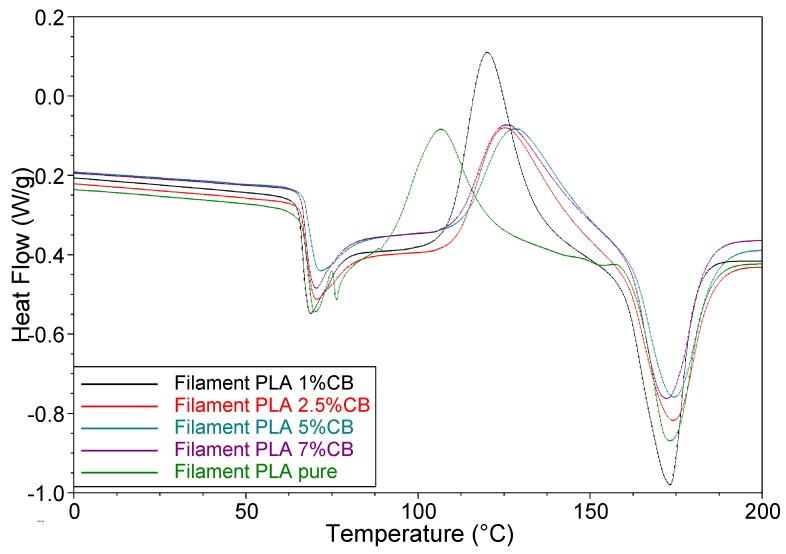
DSC curves of the unfilled PLA filament and the filled composite filaments: PLA + 1 vol.% CB; PLA + 2.5 vol.% CB; PLA + 5 vol.% CB and PLA + 7 vol.% CB. The numbers near the curves indicate the content of CB in composites.

**Figure 10 polymers-14-04022-f010:**
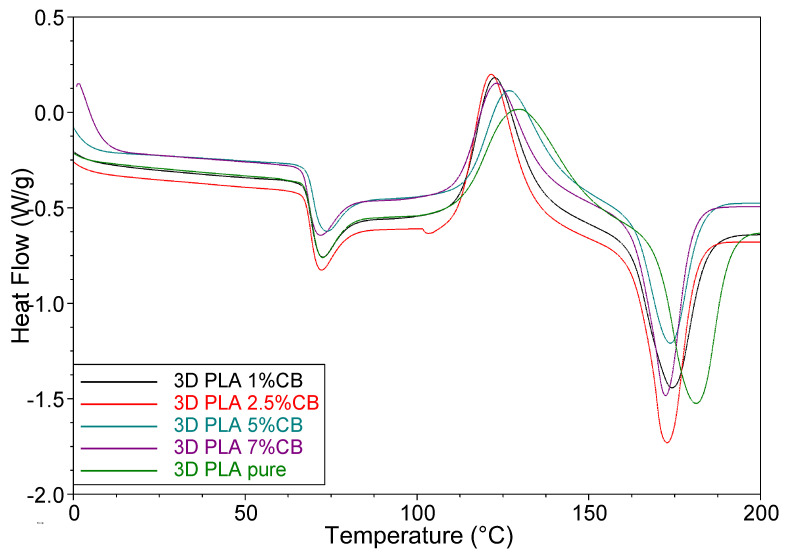
DSC curves (Ⅰ run) of the 3D-printed specimens, obtained from the composite filaments with various CB content.

**Figure 11 polymers-14-04022-f011:**
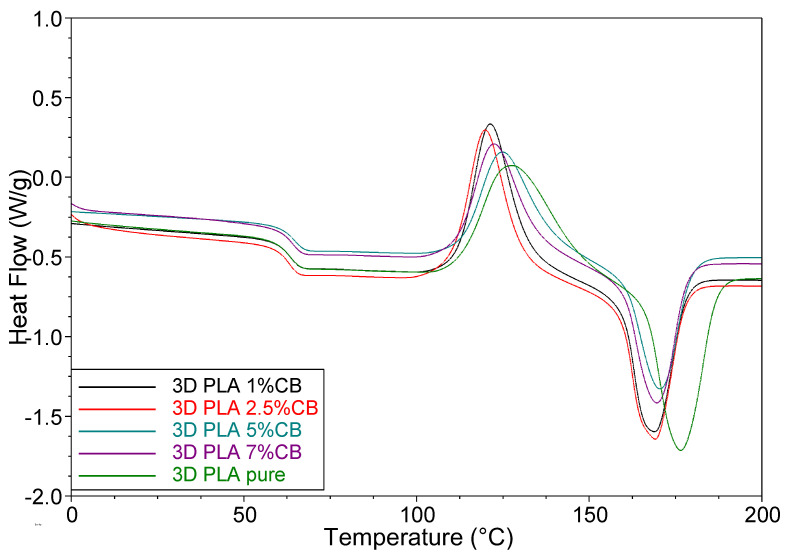
DSC curves (Ⅱ run) of the 3D-printed specimens, obtained from the composite filaments with various CB content.

**Figure 12 polymers-14-04022-f012:**
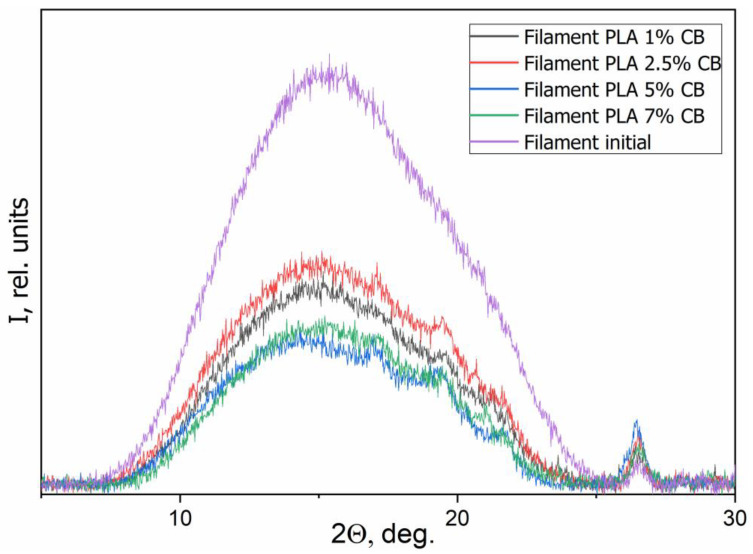
WAXS patterns of the composite filaments with various CB content *φ*.

**Figure 13 polymers-14-04022-f013:**
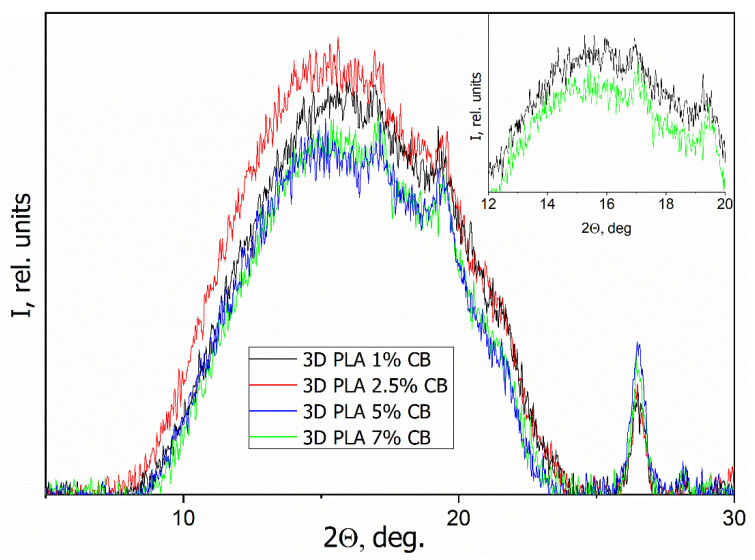
WAXS patterns of the 3D specimens, obtained from the composite filaments with various CB content *φ*, inset in the figure shows patterns for PLA/CB composites with 5 and 7 vol.% of CB.

**Figure 14 polymers-14-04022-f014:**
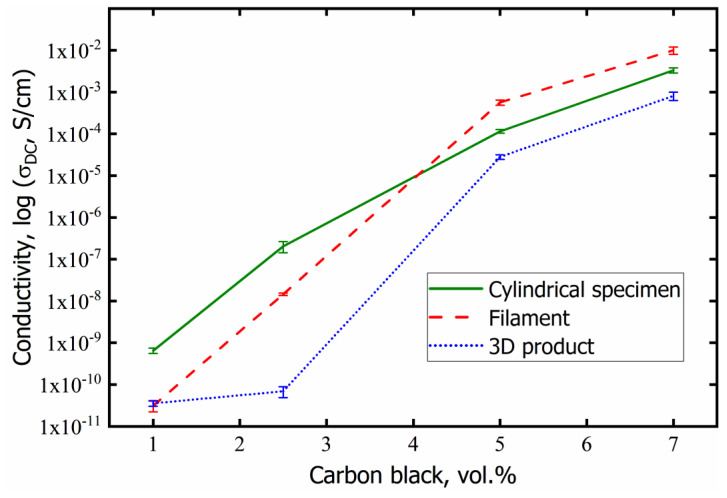
Dependences of electrical conductivity *σ_dc_* of specimens on the CB content.

**Figure 15 polymers-14-04022-f015:**
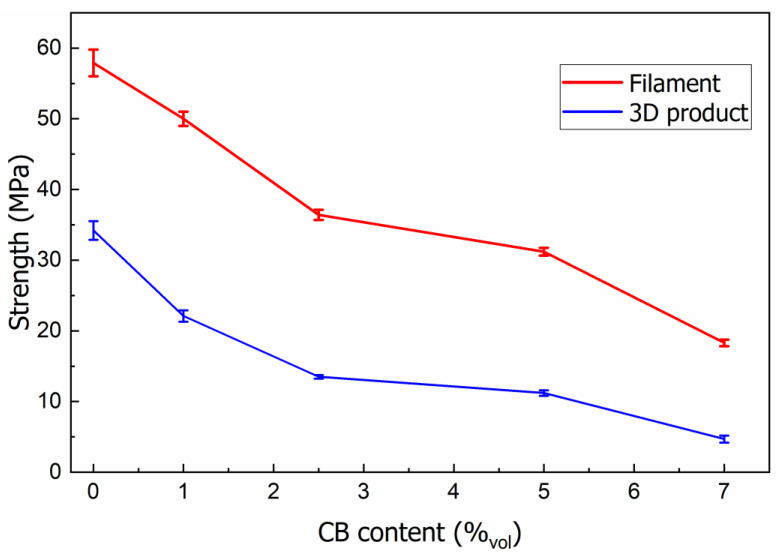
Dependences of mechanical strength of specimens on CB content.

**Table 1 polymers-14-04022-t001:** Thermal properties of the PLA/CB composites.

Filler Content *φ*, vol.%	*T_g_,* °C	*T_cc_,* °C	Δ*H_cc_,* J/g	*T_m_,* °C	Δ*H_m_,* J/g	*X_c_*, %
** *Filaments (Ⅰ run)* **						
**0**	67.8	106.8	17.9	173.1	18.5	0.68
**1**	66.7	120.2	23.7	173.2	25.0	1.57
**2.5**	67.0	125.3	20.2	173.8	21.3	1.40
**5**	67.9	128.5	17.6	174.3	19.1	1.20
**7**	67.2	126.1	17.1	173.2	19.6	3.06
** *3D specimens (Ⅰ run)* **						
**0**	69.4	129.8	40.5	181.0	42.5	2.26
**1**	69.1	122.6	35.5	174.2	37.9	2.60
**2.5**	69.1	121.6	37.2	172.7	40.8	3.97
**5**	69.8	126.9	28.8	173.7	31.8	3.47
**7**	68.3	123.2	31.4	172.3	35.4	4.78
** *3D specimens (Ⅱ run)* **						
**0**	63.3	127.4	45.9	176.5	47.5	1.66
**1**	63.6	121.3	38.7	168.9	40.4	1.87
**2.5**	63.3	119.9	39.3	169.1	40.9	1.77
**5**	64.7	124.9	33.3	170.5	34.7	1.63
**7**	64.2	122.5	34.5	169.6	36.2	2.04

## Data Availability

Data are contained within the article.

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
