# Peer review of "Polylactide/Carbon Black Segregated Composites for 3D Printing of Conductive Products"

_polymers, 2022, doi:10.3390/polym14194022_

Round 1
Reviewer 1 Report
Firstly, it is a timely effort that is accomplished by the authors to present their research work on Polylactide/Carbon black segregated composites for 3D printing of conductive products. However, there are few suggession:
1. Novelty must be clarified in a better way.
2. The sentences in introduction section where the authors wrote" the authors of [15]..." must be avoided. Instead they may write the name of corresponding author.
3. The number of references are at lower side in the introduction section and they may be increased.
4. Few methods are new to me and they must be cross checked for their standardization or was this Mamunya Geomtrical model used before using it here?
Thanks
Author Response
Response to Reviewer 1 Comments
First of all, we would like to thank you for your time and useful comments and recommendations, which have helped us to improve the manuscript.
The text below contains the answers to the comments.
1. Novelty must be clarified in a better way.
Novelty arises from the expanded Introduction, where we described already conducted studies in the field of the submitted article and their minor followed problems from a potential or economic point of view. Mostly our study focuses on the production of functional 3D printed conductive products that can be used as heating elements for joining thermoplastics, which were studied in our previous paper. Moreover, such a potential usage we have never found during the literature review and this makes our study and their goals in a way novel.
2. The sentences in introduction section where the authors wrote" the authors of [15]..." must be avoided. Instead they may write the name of corresponding author.
In the sentences in introduction section, where we have previously written "the authors of [x]..." we changed for “the name of corresponding author” and further added [21, 23, 24]. The numbers, which you have mentioned, changed because of increasing references in the article. In the other cases we used the Microsoft Word template to prepare this manuscript. There is next information: “References should be numbered in order of appearance and indicated by a numeral or numerals in square brackets—e.g., [1] or [2,3], or [4–6] and placed before the punctuation; for example [1], [1–3] or [1,3]”.
3. The number of references are at lower side in the introduction section and they may be increased.
The introduction section was expanded and more references have been added.
4. Few methods are new to me and they must be cross checked for their standardization or was this Mamunya Geometrical model used before using it here?
Mamunya’s model is widely known, it is used for years and has a good correlation between experimental results and real composites. For example, in the article doi:10.1016/j.elstat.2014.02.002, several DC electrical conductivity models have been proposed to explain the properties of composite materials, including Mamunya’s model. Using different carbon black nanocomposites and Mamunya’s model, authors obtained a good agreement with the experimental results for concentrations above the percolation critical concentration. Also we added to manuscript some references to Mamunya’s articles, which describe his model.
Additionally:
In our manuscript we have replaced figures 12 and 13 to the similar, changing only frame with legends (without shadow) for oneness.
In figure 14 we have deleted points for better visibility of error bars.
We look forward to hearing from you regarding the submission of our manuscript. We will be glad to answer any additional questions and comments you may have
Reviewer 2 Report
The authors reported an interesting work using FDM methods to print PLA/carbon black composites. The work is well designed, and the paper is well written. However, there are some flaws in the data and some missing reference. The introduction also needs to be improved. Therefore, I suggest a major revision before I can reconsider it for publication.
1. Line 105-107, . “Interventionary studies involving animals or humans, and other studies that require ethical approval, must list the authority that provided approval and the corresponding ethical approval code” is not relevant to the paper at all and is clearly copy-paste by mistake.
2. It is not clear to me what images they are in Figures 5 and 6. Are they SEM images? Please be specific.
3. Additive manufacturing/3D printing (the second paragraph in the introduction) could be introduced a bit more such as the more advantages and real-life applications.
4. Following question 3, some recent examples of using 3D printing method to add conductive filler into the polymer matrix to fabricate polymer composites (https://doi.org/10.1039/D1PY00705J; https://doi.org/10.1016/j.apmt.2017.04.003) should be cited and briefly discussed, as they are very related to this work.
5. Figure 1, the SEM images have very low quality. I suggest the authors redo them.
6. Table 1, instead of listing the data obtain from 2 run, I suggest the authors just get the average value and give the error.
7. Similar to question 6, the authors can combine the Figure 10 and 11 and give the error bar.
8. Figure 14, are the data obtain from a couple of runs or just measure for one time? I suggest the authors run at least times with two printed samples and give the error bar on the figure.
9. Figure 8a, can you label in the SEM that which part is polymer and which part is carbon black? The current one is not clear to readers.
10. In section 3.4, the authors claimed that by adding more CB, the better crystallinity. The authors directly compare the intensity from WAXS, which is not convincing, since the WAXS/XRD are bulk technics and unless it is an in-situ experiments, the authors’ conclusion cannot stand well. The intensity cannot be directly compare since the many other factors such as the measurement amount will also influence the it. I suggest the authors revise the claim.
Author Response
Response to Reviewer 2 Comments
Firstly, we would like to thank you for taking the time to do a thoughtful analysis and valuable comments and suggestions. We appreciate the informative help that you have provided. Your input will definitely help to improve this article.
The next paragraph contains the answers to the comments.
- Line 105-107, . “Interventionary studies involving animals or humans, and other studies that require ethical approval, must list the authority that provided approval and the corresponding ethical approval code” is not relevant to the paper at all and is clearly copy-paste by mistake.
We are grateful for this kind comment. It’s a mechanical mistake. When filling out the template, we didn’t delete the part of the text from it.
- It is not clear to me what images they are in Figures 5 and 6. Are they SEM images? Please be specific.
It’s an optical microscopy. We have already added this information to the article.
- Additive manufacturing/3D printing (the second paragraph in the introduction) could be introduced a bit more such as the more advantages and real-life applications.
Introduction was enhanced with additional information on real-life and possible future applications. References, which prove corresponding data, have been added.
- Following question 3, some recent examples of using 3D printing method to add conductive filler into the polymer matrix to fabricate polymer composites (https://doi.org/10.1039/D1PY00705J;https://doi.org/10.1016/j.apmt.2017.04.00) should be cited and briefly discussed, as they are very related to this work.
Reference of the second article that is relevant to the topic of submitted paper has been added to the introduction section, describing future possible application of conductive polymer composites in additive manufacturing.
- Figure 1, the SEM images have very low quality. I suggest the authors redo them.
Such images are the best made using the mentioned microscope, which we have. Unfortunately, reservations and redoing images to little quality improvement cannot be made during the 10-day period of submitting a revised article due to internal working issue with outside present microscope.
- Table 1, instead of listing the data obtain from 2 run, I suggest the authors just get the average value and give the error.
To our mind that cannot be done, since we are not able to speak about average value for different samples with different content of CB and the reader will not see the tendency, which we have established
- Similar to question 6, the authors can combine the Figure 10 and 11 and give the error bar.
To our mind the combination of the Figure 10 and Figure 11 will reduce the readability of the data of graphs, because of that we have divided them in two Figures.
- Figure 14, are the data obtain from a couple of runs or just measure for one time? I suggest the authors run at least times with two printed samples and give the error bar on the figure.
The results were obtained for several samples. As we mentioned in the methods we made measurements of three samples per one concentration of CB. The error bars have been added in the figure 14.
- Figure 8a, can you label in the SEM that which part is polymer and which part is carbon black? The current one is not clear to readers.
We agree that in the given picture is not clear to readers which part is a polymer and which part is carbon black. That’s why we change picture and have pointed CB in Figure 8a.
- In section 3.4, the authors claimed that by adding more CB, the better crystallinity. The authors directly compare the intensity from WAXS, which is not convincing, since the WAXS/XRD are bulk technics and unless it is an in-situ experiments, the authors’ conclusion cannot stand well. The intensity cannot be directly compare since the many other factors such as the measurement amount will also influence the it. I suggest the authors revise the claim.
We would like to thank reviewer for comment. However, considering experiment parameters, which were used for examinations, it should be noted that results can be compared. In the following paper, at least three different measurements were performed on three randomly selected samples, with same amount (approximately 0.5 g). Additionally, sample's rotation was applied during measurements, what allows to claim that such results can be treated as representative. Considering above, the stated sentences are correct.
Additionally:
In our manuscript we have replaced figures 12 and 13 to the similar, changing only frame with legends (without shadow) for oneness.
In figure 14 we have deleted points for better visibility of error bars.
We look forward to hearing from you regarding the submission of our manuscript. We will be glad to answer any additional question and comments you may have
Round 2
Reviewer 2 Report
Authors made some efforts but all the required experiments were not added. I will just give authors one more chance to revise the manuscript.
1. For example, my question 5 in the first review, I asked the authors to get better SEM images. The current one is with really low quality and the readers cannot get any information from it at all. In fact, the current SEM should not be published in any journal, since it is a misleading data. I do not think that limited time is a good excuse for authors to publish bad data.
2. It is quite common for the researchers to give error bar or give the average value for the same experiments with the same conditions but for different run. I asked the authors either give the error bar or average number based on their current data in my questions 6 and 7. This is just a very simple analyze but the authors just did not do it nor gave any good, solid or scientific reason to show why they did not do it.
Author Response
- Table 1, instead of listing the data obtain from 2 run, I suggest the authors just get the average value and give the error.
- Similar to question 6, the authors can combine the Figure 10 and 11 and give the error bar.
It is quite common for the researchers to give error bar or give the average value for the same experiments with the same conditions but for different run. I asked the authors either give the error bar or average number based on their current data in my questions 6 and 7. This is just a very simple analyze but the authors just did not do it nor gave any good, solid or scientific reason to show why they did not do it.
First of all, on behalf of the authors, I thank the reviewer for useful comments. With regard to remarks 6 and 7, I would like to clarify the situation with the presence in the article and, in particular, in Table 1 of the data obtained from run I and run II. In the DSC method, it is common practice to make two runs. This is done not for obtaining the average data or for reducing the measurement error, but is caused the fact that during the first run the DSC curve additionally includes thermal effects, which are due to the processing conditions of the composite specimen. It is generally accepted that during the first run and subsequent cooling at the same rate, the effect of processing is eliminated, and during the second run the DSC curve reflects only the thermal effects associated with the polymer structure. For this reason, only the second run data is often given in the literature, without even indicating that this is the second run.
In our case, we provide first and second runs data for 3D specimens. We wanted to see what structural changes the printing process produces. As can be seen from Table 1, there are differences between run I and run II, in particular the glass transition temperature Tg is lower for the second run. The most significant changes are for the degree of crystallinity Xc, which is approximately two times lower in the second run (as can be seen from Table 1, due to an increase in the crystallization heat DHcc). These data are analyzed and discussed in the text of the article (two sentences are included in the text that clarify the effect of the second run). In this regard, we present separately Fig. 10 and Fig. 11, which show the DSC curves after the first and the second run.
As for the measurement error, it is not customary to put an error bar on DSC curves. The fact is that the DSC devices manufactured by different companies have approximately the same measurement accuracy, which is ±0.01 0C in temperature and ±0.001 W/g in heat flow. Researchers who work with the DSC method know these values, so the error bar not necessary to plot on the DSC curves. Our DSC data are presented in the style in which they are usually given in the literature.
- Figure 1, the SEM images have very low quality. I suggest the authors redo them.
For example, my question 5 in the first review, I asked the authors to get better SEM images. The current one is with really low quality and the readers cannot get any information from it at all. In fact, the current SEM should not be published in any journal, since it is a misleading data. I do not think that limited time is a good excuse for authors to publish bad data.
We redone SEM images and changed them for ones in higher quality.
Round 3
Reviewer 2 Report
fine